# Rule-Based Detection of False Data Injections Attacks against Optimal Power Flow in Power Systems

**DOI:** 10.3390/s21072478

**Published:** 2021-04-02

**Authors:** Sani Umar, Muhamad Felemban

**Affiliations:** Computer Engineering Department, King Fahd University of Petroleum and Minerals, Dhahran 31261, Saudi Arabia; g201706410@kfupm.edu.sa

**Keywords:** cyber-security, intrusion detection system, smart grid

## Abstract

Cyber-security of modern power systems has captured a significant interest. The vulnerabilities in the cyber infrastructure of the power systems provide an avenue for adversaries to launch cyber attacks. An example of such cyber attacks is False Data Injection Attacks (FDIA). The main contribution of this paper is to analyze the impact of FDIA on the cost of power generation and the physical component of the power systems. Furthermore, We introduce a new FDIA strategy that intends to maximize the cost of power generation. The viability of the attack is shown using simulations on the standard IEEE bus systems using the MATPOWER MATLAB package. We used the genetic algorithm (GA), simulated annealing (SA) algorithm, tabu search (TS), and particle swarm optimization (PSO) to find the suitable attack targets and execute FDIA in the power systems. The proposed FDIA increases the generation cost by up to 15.6%, 45.1%, 60.12%, and 74.02% on the 6-bus, 9-bus, 30-bus, and 118-bus systems, respectively. Finally, a rule-based FDIA detection and prevention mechanism is proposed to mitigate such attacks on power systems.

## 1. Introduction

Smart power systems are complicated cyber-physical systems. An example of such a system is the modern power systems, which consist of physical infrastructure and cyber-infrastructure. The physical infrastructure is responsible for generating, transmitting, and distributing electric power. On the other hand, the cyber infrastructure includes the telemetry and the communication equipment connected to the power systems. The cyber components help in enhancing decision-making and monitoring of the power systems. Furthermore, cyber components, such as meters, are used to measure parameters, such as the load demand of a particular node (i.e., bus), within the power system network. The energy management system is composed of network topology, economic dispatch calculations, Power Systems State Estimator (PSSE), Automatic Generation Control (AGC), contingency analysis, and Optimal Power Flow (OPF). Moreover, the energy management system is responsible for the controlling, monitoring, and optimizing the performance of the performance of the power systems [1].

Recently, the cyber-security of power systems has captured a significant interest. The analysis of electric grid security under terrorist threat is presented in Reference [2]. A model of load redistribution attacks is studied in Reference [3]. In Reference [4], Cyber-security of smart grid infrastructure is studied. Similarly, the cyber-security and privacy issues in smart grids are studied in Reference [5]. A survey on the cyber-security of smart grid is presented in References [6,7]. Authors in Reference [8] use sparse optimization to detect false data injection attacks in power systems. The effect of FDIA and contingency is studied in References [9,10,11]. Authors in Reference [12] presented an FDIA identification mechanism in smart grids. This is because the world has seen a surge in cyber-attacks on power systems. Example of such attacks include the USA generator explosion in 2007 [13], the Ukraine blackout in 2015 [14], and the Turkey oil explosion in 2008 [13]. Table 1 presents examples of some major attacks that happened in the energy industry. In these attacks, an adversary leverages the vulnerabilities in the cyber infrastructure to attack the power systems. This leads to catastrophic consequences that impact the physical components of the power systems.

The proposed attacks and defensive strategies can be implemented in real power systems. The adversary can compromise measurement equipment physically, such as electric meters and sensors. The attacker can intercept data packets in the power system network while they are transferred to the control center. In addition, the attacker can modify the database of the control center through unauthorized access. This unauthorized falsification of data can mislead the operation of the power systems which leads to an increase in the generation cost. The proposed defensive mechanism can be implemented on the real power systems control centers. Subsequently, This can enhance decision-making and help in detecting and preventing false data injection attacks against real power systems. In addition, it is important to note that the standard IEEE bus systems are a simple approximation of the real power systems. Hence, we used them to evaluate the performance of the attacks and defensive mechanism.

The main contribution of this paper is the detailed analysis of the impact of False Data Injection Attack (FDIA) on the power systems. In particular, we proposed an Advanced Persistence Threat (APT)-based FDIA that intends to maximize the negative impact on the cost of power generation and the physical component of the power systems. The novel APT-based FDIA strategy allows attackers to attack multiple nodes to maximize the impacts of the attacks on power systems. We formulated the attack model as an optimization problem with the objective to maximize the cost of power generation. The main difference between our proposed APT-based FDIA and the existing FDIA from the literature is that we usee metaheuristics techniques, namely genetic algorithm (GA), simulated annealing (SA), and particle swarm optimization (PSO), to conduct the attack. Then, we evaluate the attack by running simulations on the standard IEEE bus systems using MATPOWER in MATLAB [15]. In addition, we proposed a rule-based detection mechanism to mitigate FDIA in power systems. We used accuracy metrics to evaluate the performance of the developed rule-based FDIA detection and prevention system.

The remaining sections of this paper are as follow. Section 2 presents the related work and highlights the background concept of this thesis. Section 3 presents the proposed attack strategy. Section 4 presents the proposed FDIA detection system. Section 5 discusses our findings. Finally, the conclusion and future work are presented in Section 6.

## 2. Background and Related Work

This section highlights the related work and background of the problem. First, we present a classification of attacks based on the attack execution (Figure 1). Then, we present state-of-the-art mechanisms to detect data integrity attacks in power systems. Next, we present the model of the system under investigation. Using the system model, we present a mathematical model of the OPF problem, which is in the form of a combinatorial optimization problem.

### 2.1. Attacks on Power Systems

As a form of a cyber-physical system, power system is vulnerable to different types of attacks. Attacks on cyber-physical systems can be classified into cyber-based, communication-based, network-based, or physical-based attacks [13]. There are various work in the literature to investigate different cyber attacks on power systems. Yang et al. [1] investigate the data integrity attack as a security threat to the OPF in power systems. A critical attack vector is modeled such that the attacker can execute attacks with minimum effort in terms of the amount of information to falsify and the number of nodes to compromise. Suna et al. [16] present a survey on the state-of-the-art of pertinent issues related to cyber-security in power systems. A survey of communication-based cyber-security issues is presented in Reference [17]. Authors in References [18,19] present a survey on cyber attacks and defensive strategy in power systems. Cyber-based attacks are executed through the cyber layer of a system. Cyber-based FDIA occurs when an attacker injects false information into the cyber components of the system [20]. Code manipulation occurs when an adversary manipulates the code implementation of the system software. Malware injection occurs when a virus is injected into the system. All the aforementioned attacks are considered cyber-based attacks. The reason is that they can be executed through the cyber layer of the power systems.

Network-based attacks are executed through unauthorized access to the network. In such attacks, the system software and the physical communication links are not affected. Denial of service (DoS) attack is considered network-based attacks. In a DoS attack, an adversary floods the network with fake request packets that render the network inaccessible to legitimate activities. Other network-based attacks include packet sniffing attack and man in the middle attack [21]. Moreover, FDIA can be executed through the network layer of a system. False information can be injected into the packets that traverse across the network. Other types of attacks that fall under this category include packet sniffing, man in the middle, and black hole attacks.

Physical-based threats are types of attacks where an adversary gains physical access to the system and damage its components. This can disturb the normal operation of the target systems. FDIA in which an adversary has physically accessed, compromised devices, and inject false information into their input is considered a physical-based attack. Other kinds of attacks in this category include Emission Security (Emsec) attack and the use of electromagnetic pulse to physically damage systems [22]. Communication-based attacks are types of attacks that impact communication links. Message replay and channel jamming attacks are categorized under communication-based attacks [23]. This is because they can be executed on the communication links to impose negative consequences on the normal operation of the target systems [13].

### 2.2. Detection of False Data Injection Attacks

Some studies attempt to propose various mechanisms to detect data integrity attacks in power systems [24,25,26,27,28,29,30,31]. The detection algorithms can be data-driven-based [32,33,34,35] or model-based [1,36,37]. In data-driven-based detection algorithms, machine learning algorithms are utilized to processed data collected from the power systems to detect false data injection attacks.

Rahman et al. [37], present a formal model of undetected data integrity attack on state estimation that can deviate from the bad data detection module and compromise the solution provided by the optimal power flow module in the smart grid. The proposed model is utilized in investigating and verifying the influence of false data injection attacks on the optimal power flow. The proposed attack model is based on the formalization of certain attributes. These attributes include the adversary’s knowledge of the network, adversary’s resource constraints, and accessibility to the resources and suitable attack targets. Moreover, an example case study was carried out on the IEEE 5-bus system. This is to delineate the proposed model. It was shown that the feasible attack increases the cost of operation of the OPF by up to two percent (2%).

Yang et al. [1] investigate the data integrity attack as a kind of a cyber-security threat to the OPF in power systems. A critical attack vector is modeled in such a way that the attacker can execute attacks with minimum effort. From the adversary’s perspective, the target buses to compromise are selected by examining the difference between the real transmission power and line capacity limitation. Therefore, a transmission line with the smallest difference is selected as the target line, and all nodes connected to that particular line are selected as the target nodes to compromise. An efficient defensive strategy is presented. This includes both detection-based strategy and protection-based. The detection-based strategy detects the presence of false measurements. The protection-based strategy protects against data integrity attacks by deploying smart meters and sensors.

Esmalifalak et al. [32] proposed a machine learning-based detection mechanism against stealthy false data injection attack in power systems. The proposed technique utilizes a labeled data set collected from the operation of the smart grid. The dataset is used to train a supervised distributed support vector machine. It was shown that the normal operation of the smart grid can be strategically distinguished from its operation under false data injection attacks. Furthermore, the three commonly used supervised machine learning classifiers (i.e., k-nearest neighbor (kNN), extended nearest neighbor (ENN), and support vector machine (SVM)) are utilized in Reference [33] to detect false data injection attack in power systems. All three classifiers yield more than 80% accuracy on the IEEE standard 30-bus system. The 30-bus system was chosen as a benchmark test case. It was shown that supervised learning-based detectors are an effective way to combat false data injection attacks in power systems. Real-time detection of false data injection attacks using Artificial Neural Networks (ANNs) is proposed by Erik et. al. [34]. The proposed detection techniques is capable of detecting not only data integrity attacks but can also detect sensor failures and replay attacks. An SVM-based detection technique is used to compare the performance of the proposed real-time detection mechanism. Experimental results showed that the proposed detection techniques achieved 99% accuracy. However, the authors failed to narrate the amount of training data required for the proposed learning-based technique to detect data integrity attacks.

Authors in Reference [38] studied false data injection attacks on security-constrained OPF. A bi-level optimization problem model of a false data injection attack is utilized to figure out the minimum number of sensors required to compromise. Moreover, vulnerable critical generators and transmission lines are identified. The standard IEEE 14 and 30 test cases are used to investigate the vulnerability of the power system against false data injection attacks. Apart from the detection techniques of false data injection attack proposed in the literature, various prevention-based techniques for false data injection attacks have also been investigated. Abdallah et. al. [39] proposed an efficient prevention-based technique for false data injection attacks in power system. The proposed technique aims to prevent the occurrence of false data injection attacks by assuring the availability and integrity of power systems measurements from all units in the network. Simulation results indicate that the proposed prevention technique has negligible computational complexity and communication overhead. Manandhar et al. [40] discuss various drawbacks in the security of power systems, considering the telemetry and communication equipment, including sensors and actuators.

In literature [8], the authors viewed FDIA as a matrix separation problem. They examine the sparse nature of FDIA and the dimensionality of measurements of the power systems. A novel FDIA detection technique is proposed. The technique can detect a malicious attack and identify a normal system state. This is done by using the factorization of the low-rank matrix and minimization of the nuclear norm. The authors conducted a series of numerical simulations on both real data and synthetic data to validate the effectiveness of the proposed detection technique. However, the authors do not analyze the impact of FDIA especially on the cost of power generation and the impact on the physical components of the power systems.

Lei et al. [41] proposed a Deep Belief Networks (DBN)-based detection of FDIA in the smart grid. The authors used an unsupervised machine learning approach to train the detection technique in order to differentiate between the malicious system state and the normal system state. Backpropagation is used as an algorithm to fine-tune the parameters of the model and propagate the errors. To evaluate the effectiveness of the proposed detection mechanism, a series of simulations were carried out on the IEEE standard systems. The authors found that their proposed FDIA detection performed better than support vector machine (SVM)–based FDIA detection.

Zhang et al. [42] studied the security issues of dynamic state estimators when the communication and telemetry equipment are compromised by adversaries using FDIA. The authors proposed a sufficient conditions such that the attack can be executed with stealthily. Moreover, the energy of the attacks stealthiness were introduced. Finally, the superiority of the attacks were illustrated on the standard IEEE 6-bus systems.

In addition, Lu et al. [43] investigate FDIA on power systems state estimator in the presence of telemetry and communication equipment failure. The power system under investigation is equipped with the bad data detection mechanism. An undetectable FDIA is modeled to worsen the performance of the power system state estimation. Finally, the effectiveness of the attacks model is evaluated using the IEEE standard 5-bus, 9-bus, and 30-bus systems.

Sayghe et al. [44] presented a survey of machine learning methods to detect FDIA in power systems. The authors presented comprehensive background information on FDIA, the impact of FDIA on power systems, and the FDIA defense mechanisms that utilize the machine learning approach to detect FDIA in power systems. The detection mechanism includes protecting the least number sets of meter, game theory-based defense, cryptographic-based defense, topology, and proactive-based defense against FDIA. Besides, the limitation of those detection mechanics include intensive computation, adversary ability to predict new systems configuration, protection only limited set measurements, and modeling challenges. Ashrafuzzaman et al. [45] proposed a data-driven-based mechanism to detect FDIA using an ensemble-based machine learning approach. Several classifiers are used to detect FDIA in the proposed approach. In addition, two ensembles are used in which one utilizes supervised classifiers and the other uses unsupervised classifiers. The proposed detection mechanism is evaluated on the IEEE standard 14-bus system using simulated data. Their results show that the ensembles are better than the individual classifies for the unsupervised models.

### 2.3. Bus System Model

Figure 2 depicts an example of the physical infrastructure, known as the IEEE 5-Bus system. Examples of other IEEE standard systems include; 9-Bus, 14-Bus, 24-Bus, 30-Bus system, 39-Bus, 57-Bus, and 118-Bus [15]. A typical power system consists of four components [46]: power generators, telemetry equipment, busses, and transmission lines. The generators ($g_{1}$, $g_{2}$, $g_{3}$, $g_{4}$, and $g_{5}$) are responsible for power generation. On the other hand, the telemetry equipment ($Pd_{2}$, $Pd_{3}$, and $Pd_{4}$) are devices, such as meters and sensors that collect power systems measurements. The buses ($Bus_{1},Bus_{2},Bus_{3},Bus_{4},$ and $Bus_{5}$) are the connecting point between transmission lines and generators. Transmission lines are the lines connecting two buses, which are responsible for the transmission of power from one location to another, across a country or region.

For example, the 5-bus system, depicted in Figure 2, consists of two generators ($g_{1}$ and $g_{2}$) attached to bus number 1, one generator ($g_{3}$) attached to bus number 3, one generator ($g_{4}$) attached to bus number 4, and one generator ($g_{5}$) attached to bus number 5. A residential load which mainly consists of heating, lighting, and cooling is connected to bus numbers 2, 3, and 4 which represent the transmission substations. The loads are represented by the variables $P_{d2}$, $P_{d3}$, and $P_{d4}$, respectively. All buses are connected by transmission lines to form a power system network and enable power flow between them. In the system model, it is assumed that all generators are secure and cannot be attacked by an adversary. That is, false data cannot be injected into the measurement related to the generator output. Therefore, the only measurement available to the attacker is the measurement related to the load demand in the power system network. Hence, the attacker can only deviate the optimal generator output by falsifying the load demand of certain nodes in the power systems network.

### 2.4. The Energy Management System

In power systems, the energy management system is responsible for the control, monitoring, and optimization of the performance of the power systems [1]. As depicted in Figure 3, the energy management system is composed of network topology, economic dispatch calculations, Power Systems State Estimator (PSSE), Automatic Generation Control (AGC), contingency analysis, and Optimal Power Flow (OPF).

Network topology is the representation of the entire network. The network topology is composed of the connectivity of the buses and how the loads and generators are connected to the buses within the network. The economic dispatch calculations module is used to determine the target power output of all generators in the network. PSSE is one of the critical components of power systems. The PSSE estimates the state of power systems based on data collected by the sensors from the power systems network. The AGC module is used to adjust the power generation output of all generators so that the change in the load demand is satisfied. This is because there is a need to meet the power flow constraint that requires that the generated power and load demand of the power systems is balanced. The contingency analysis module is used to identify unexpected failures of power systems components, such as failure of transmission lines or generators. In the following section, we discuss the OPF component.

### 2.5. The Optimal Power Flow Problem

OPF depends on the output of the state estimation module to accurately determine the optimal operating level of the generators. This is required for automatic generation control. Recent research efforts have shown the vulnerability of the state estimation to cyber-attacks [47]. Consequently, an attacker can inject false information into the telemetered measurements that corrupt the output of the state estimation and consequently mislead the OPF mechanism.

An OPF mechanism is utilized in determining the optimal operating level of generators. The objective of the OPF mechanism is to reduce the overall cost of generation while meeting the load demand, generations, and transmission constraints. These constraints include the capacity limitation of the generators and the transmission lines. Since the OPF mechanism depends on power systems measurements from meters and sensors, the compromised false measurements can mislead the operation of the OPF mechanism in determining the optimal operating level of the power systems.

In this paper, we consider Direct Current (DC)-OPF, where a unit value is considered as the voltage value. The OPF problem can be mathematically formulated as follows. Consider a power system with a set of buses $\Omega_{net}$, a set of transmission lines $\Omega_{L}$, and a set of generation points $\Omega_{g}$. The OPF Problem formulation consists of the cost function, inequality and equality constraints. The main objective of OPF is to minimize the operational cost of the generators in the network. The operational cost is modeled as the cost of fuel consumption by the power generators. In particular, the cost of a particular generator *i* is modeled as a quadratic cost function provided in Equation (Equation 1), where $a_{i}$, $b_{i}$, and $c_{i}$ are the cost coefficients related to the fuel cost [48]. The cost coefficients are determined by the properties of the generator. In this paper, we adopted the standard cost coefficients from MATPOWER MATLAB package [15]. Moreover, the inequality constraints of the formulation of the OPF optimization problem includes the capacity limitation of the transmission lines and generators, as shown in Equations (2) and (3), respectively. $P_{ij}$ represents the power flow from bus i to bus j and $P_{g}$ represents the generator output. Furthermore, the OPF optimization problem formulation includes equality constraints as shown in Equation (Equation 4). $B_{ij}$ is the line admittance between bus *i* and *j*, and $\delta_{i}-\delta_{j}$ is the difference between the voltage phase angle at node (bus) i and j. In addition, as shown in Equation (Equation 5), the equality constraints include the equation that balances the real power inflow and outflow at each buses in the power systems network. Table 2 summarizes the modeling parameters. The following is the formulation of the OPF optimization problem according to Reference [46]:(1)$$\begin{array}{cc} \min & \;F\left(P_{g_{i}}\right)=\sum_{i\in \Omega_{g}}^{n}a_{i}{P_{g_{i}}}^{2}+b_{i}P_{g_{i}}+c_{i},\\ s.t. \end{array}$$(2)$$\;P_{g_{i}}^{min}\leq P_{g_{i}}\leq P_{g_{i}}^{max},$$(3)$$\;|P_{ij}|\leq P_{ij}^{max},\;\;\forall i,j\in \Omega_{net},$$(4)$$\;P_{ij}=B_{ij}*\left(\delta_{i}-\delta_{j}\right),\;\;\forall i,j\in \Omega_{net},$$(5)$$\;P_{g_{i}}-P_{d_{i}}-\sum_{j\in \Omega_{net}}P_{ij}=0,\;\;\forall i\in \Omega_{net}.$$

### 2.6. Metaheuristics

Metaheuristics are designed to solve optimization problems in the case where other methods are either inefficient, ineffective or failed. Metaheuristics are considered one of the most practical methods for solving many optimization problems, and it is applies to the real-world problems that have the nature of combinatorial problems. Metaheuristics can be single solution-based, such as simulated annealing, population-based, such as genetic algorithm and tabu search (TS), or swarm intelligence, such as particle swarm optimization (PSO).

#### 2.6.1. Simulated Annealing

To make a perfect and strong metallic object, a controlled metal cooling procedure is followed, this procedure is known as annealing. It refers to the heating of the metal to a very high temperature and follows a controlled cooling process by reducing the temperature is steps. In every step, the temperature is maintained for the metal to attain thermal equilibrium [49]. The simulated annealing metaheuristics proposed in the year 1983 is based on the analogy of simulating the annealing of metal to solve optimization problems. The algorithm mimics the process of crystallizing a melted metal into a solid. The same analogy is applied to computer science algorithms to solve problems [50,51].

#### 2.6.2. Genetic Algorithm

A genetic algorithm is a heuristic search that is used to solve a wide range of optimization problems. This flexibility makes the algorithm applicable to several optimization problems in practice. The basics of genetic algorithm is the evolution [52]. The success of different varieties of species is a good reason to believe in the power of evolution. Species can develop a structure that enables them to compete and survive in different types of habitat. The genetic algorithm is based on a group of individual candidate solutions that constitute a solution to the optimization problem needed to be solved. A fitness function is used to know the quality of the candidate solutions. In this paper, we used the power generation cost function for the OPF mechanism to determine the fitness of the candidate solutions. The genetic algorithm goes from one generation to another as the number of iterations increases. In every iteration, only the fittest candidate solutions survived after going through genetic operators of mutation and crossover. This allows the algorithm to approach the best solution as the number of iterations increases.

#### 2.6.3. Particle Swarm Optimization

In particle swarm optimization, particles are distributed in the search space to determine the best solution of an optimization problem. Each particle evaluates the objective function at its current position. Then, each particle decide its movement in the search space by comparing its current position and its best fitness position from the past and the group’s (swamp) best fitness. In every iteration, the swamps moves closer to the best solution to the objective function [53].

#### 2.6.4. Tabu Search Algorithm

Tabu search Algorithm is another metaheuristics technique that is used to solve combinatorial optimization problems. The algorithm is proven to be effective in solving such a kind of optimization problems [54].

## 3. Advanced Persistent Threat-Based False Data Injection Attacks

There are three ways for an attacker to manipulate measurements in FDIA [14]. The attackers can compromise measurement equipment physically. The attacker can intercept data packets in the power system network while they are transferred to the control center. In addition, the attacker can modify the database of the control center through unauthorized access. The attacker cannot launch an attack on the power systems generators, because they are assumed to be secure [1]. That is, measurements related to the power generators are not available to the attacker. Therefore, only the load demand is available for the attacker to falsify. The attacker can instigate a collusive strategy to manipulate at least two nodes simultaneously. The attacker reduces load demand of one node and increases the load demand of another node by the same amount, such that the total load demand on the system before the attack is equal to the total load demand on the system after the attack in order to reserves the OPF constraints. This is because the system will be unstable if the OPF constraints are not satisfied. The currently proposed FDIA models are not effective because there is no consideration of imposing maximum damage to the system. Therefore, another approach is to use the APT strategy to execute FDIA. In APT strategy, an adversary sustains prolonged access to the network and injects false data stealthily in order to impose maximum damage to the system. Moreover, the attacker can only falsify power demand information. The attacker has information related to the actual power flow and power limitations on the transmission lines of the power system network. The attacker intends to launch an attack to maximize the cost of power generation in power systems using the APT strategy.

### 3.1. Methodology

This section represent the methodology of this paper. The section is divided to two sections. First, from the attackers perspective, we highlighted the FDIA attack plan. Second, from the defensive perspective we highlighted the FDIA mitigation. In Section 5, we compared our results with the results presented in Reference [1] from the literature. We selected this particular paper because they have achieved better results compared to the other papers from the literature, specifically, looking from the attackers perspective with the aim of imposing maximum damage to the system in terms of the increase in generation cost. To the best of our knowledge, the authors have achieved the highest increase in generation cost when the standard IEEE bus systems are under FDIA attacks.

#### 3.1.1. False Data Injection Attacks Plan

In this paper, we analyze the impact of FDIA on the power systems in terms of the generation cost and the performance of power systems generators. Then, we introduce a novel APT-based attack strategy with the objective to maximize the operation cost of power generation. We model the attack strategy as an optimization problem, and we use metaheuristics to solve the formulated optimization problem. In particular, four metaheuristics are used, namely simulated annealing, genetic algorithm, particle swarm optimization, and tabu search, to solve the optimization problem and obtain the attack plan. The reason behind using metaheuristics is because they are considered one of the most practical methods for solving many optimization problems, and it is applies to the real-world problems that have the nature of combinatorial optimization problems.

The output of the solved optimization problem are the set of targets and the load demand to falsify. Figure 4 highlight a simple flowchart of the attack plan. First, we start with the FDIA Modeling, the output of the FDIA model is the formulated optimization problem. Next, the formulated optimization problem is solved using metaheuristics. The output of the solution is the attack targets and the amount of load demand that is feasible to falsify. This provides the attack plan for the attacker to be able to execute FDIA in power systems to maximize the impact of the attack on the power systems. Since the problem is formulated as an optimization problem, we need to have a mechanism to know how good is the solution in order to decide whether to accept the solution or not. In simulated annealing (SA), starting from the maximum temperature ($T_{max}$) to minimum temperature ($T_{min}$), a scenario to determine the suitable attack target is formulated as follows. First, we find the initial cost $\left(C_{init}\right)$ of power generation without falsifying any load demand.

To change the state of the system from one state to another, an arbitrary transmission line is selected and falsify the load demand of the buses at the two ends of the transmission line. Then, the difference between the initial cost and the current cost ($\Delta C_{i,j}$) is calculated. If the difference is positive, we accept the new systems state, else we accept the new systems state only if $exp(\frac{-\Delta C_{i,j}}{T})$ is greater than rand(0,1). The rand(0,1) gives a uniformly distributed random number on the interval of zero to one. This process is repeated until the system state with the maximum cost is found. The simulated annealing algorithm is similar to the metropolis hasting algorithm that combines both the notion of exploring and exploiting. Exploring is the idea of visiting more search space with the hope that the algorithm will get a better solution. However, the notion of exploiting is that the algorithm is always trying to find a global optimum as quickly as possible, this can lead it to stuck at a local optimum. That is, during the hill-climbing, the algorithm always wanted to exploit. Combining the two notions of exploring and exploiting means that the algorithm does not always improve, sometimes the algorithm diverge from the current system state with the hope of finding a better system state in next iteration.

To solve the formulated optimization using a genetic algorithm, we follow the following steps. The first step starts with a random selection of the initial set of solutions. That is, selecting arbitrary transmission lines to attack and selecting arbitrary load demand value to falsify. This selected initial set of solutions are called “chromosomes”, and this first step is also known as initializing population. The population is the sets of all chromosomes. That is, the set of all possible attack targets and the possible load demand values to falsify. A random uniform distribution function is used to generate the initial possible values of load demand to falsify and the possible attack targets.

In the second step, the value of the objective function of the chromosomes are calculated. This is also known as the fitness value. This step is called “selection”. This is because only the fittest chromosomes are chosen from the initial population generated in step 1. Hence, subsequent operations are applied to the selected fittest chromosomes in order to get better results.

The next step is known as “crossover’”. Here, the chromosomes are represented in terms of genes. This is done by changing the values of the chromosomes into binary. That is, the values are expressed in only zeros (0) and ones (1). As an example, the binary representation of a decimal number 8 is 1000. Moreover, crossover simply means a change in a single gene or group of genes of parents chromosomes to reproduce offspring chromosomes. This process is also known the mate of two parent chromosomes.

The next step is known as “mutation”. In this step, value of genes are altered by replacing zeros (0) with ones (1). For example, if the offspring chromosome is 1100, it becomes 0011 after the process of mutation. After the process of mutation, chromosomes in binary values are all converted to the decimal form. Then, fitness are calculated accordingly until the chromosomes produces the target fitness (the highest cost of generation).

PSO algorithm is also used to solve the formulated optimization problem. It is a swarm intelligence-based metaheuristics. A population, called swarms, of particles (candidate solutions) are used. In simple words, PSO algorithms mimic a group of birds flying and searching for food in an area known as a search space. Therefore, the algorithm is inspired by the social behavior of birds. Each candidate solution changes its position and velocity in order to move toward the optimal solution of the problem. The movement of the candidate solutions is influenced by the group’s best position and the individual’s best-known position. This guides the movement of individual particle towards the best-known position in the search space which will lead to the optimal solution. The search space can be large or small depending on the bus system under investigation. It also represents the range in which the algorithm calculates the optimal values. In this paper, each particle represents a candidate solution of a target transmission line to attack and the value of load demand to falsify. This candidate solution changes at random and moves toward a value that provides the maximum increase in generation cost in the power systems.

Another metaheuristic, called tabu search, is used to solve the formulated optimization problem. Tabu search are widely used to solve combinatorial optimization problems. This metaheuristic is a form of neighborhood search approach. It guides a local search to explore the whole solution search space beyond just local optimum. This is being done by the use of a list, called tabu list. A flexible storage is used to restrict the next iteration solution choice to some subset of neighbors of the current solutions. In this paper, the instances of the tabu list are the set of the target transmission lines and the set of the suitable load demand value to falsify that incur the highest increase in the generation cost of the power systems.

Finally, the output of the solved optimization problem (i.e., the set of target transmission lines and the set of load demand values to falsify) are used to execute the FDIA. The attacker falsify the load demand of the targets. This causes highest increase in generation cost of the power systems.

#### 3.1.2. FDIA Mitigation

As explained in Section 3, a rule-based detection and prevention mechanism is developed to metigate the FDIAs with primary and secondary detection features. In addition, we used accuracy metrics to evaluate the performance of the detection and prevention mechanism. The accuracy metrics are the True Positive (TP) rate, True Negative (TN) rate, False Positive (FP) rate, and False Negative (PN) rate. The results of the accuracy metrics are compared with the results presented in Reference [1].

### 3.2. APT-Based Attack Strategy

The proposed APT-based attack strategy addresses the following questions. Which set of transmission lines to attack? In which order should the transmission lines be attacked? What values of load demand on the nodes connecting the target transmission lines are feasible to falsify? The problem of finding the attack targets in power systems can be viewed as a search problem. There are different combinations of buses to attack and load demand values to falsify, each of which will impose different impacts on the power systems. The search space depends on the bus system network topology. We formulate the attack strategy as the following.

#### 3.2.1. Initialization

We assume that the losses in the transmission lines are all zero. Therefore, the sum of the active power output of all generators is equals to the sum of the load active power of all nodes in the power systems network [1]. The objective of our attack is to maximize the cost of power generation in the power system. Therefore, we use Equation (Equation 1) as an objective function for our attack strategy. We denote the set of the attacked transmission line as $T_{Lx}\in \Omega_{L}$. Furthermore, we denote the false load demand value for the attacked transmission lines as $\Delta T_{Lx}$.

#### 3.2.2. Optimization Problem

We model the APT-based attack strategy as an optimization problem with the objective to maximize the cost function modeled in Equation (Equation 6). The outcomes of the optimization problem are the set $T_{Lx}$ and $\Delta T_{Lx}$. The problem is formulated is the following.
(6)$$\begin{array}{cc} \max_{\Delta T_{Lx},T_{Lx}} & F\left(P_{g_{i}}\right),\\ s.t. & \end{array}$$
(7)$$|T_{Lx}|\leq |\alpha *\Omega_{L}|,\alpha \leq 1,$$
(8)$$\Delta T_{Lx}\leq P_{d_{i}}^{max},$$
(9)$$P_{g_{i}}^{min}\leq P_{g_{i}}\leq P_{g_{i}}^{max},$$
(10)$$|P_{ij}|\leq P_{ij}^{max},\;\;\forall i,j\in \Omega_{net},$$
(11)$$P_{ij}=B_{ij}*\left(\delta_{i}-\delta_{j}\right),\;\;\forall i,j\in \Omega_{net},$$
(12)$$P_{g_{i}}-P_{d_{i}}-\sum_{j\in \Omega_{net}}P_{ij}=0,\;\;\forall i\in \Omega_{net}.$$

This optimization problem is similar to the problem modeled by Equations (1)–(5). However, two inequality constraints have been added. In particular, Equation (Equation 7) bounds the number of attacked transmission lines using a defined parameter α. Equation (Equation 8) bounds the amount load demand the attacker can falsify. To compute $P_{d_{i}}^{max}$, we find the set of values of load demands that increases the power flow on each transmission line to its maximum. The remaining constraints are used to check that the system satisfies the constraints in Equations (2)–(5).

FDIA can be executed in a naive way by falsifying the load demand of randomly selected nodes without consideration of imposing maximum damage to the system. In this paper, we use an APT strategy [55] to execute FDIA to impose maximum damage to the system. Algorithm 1 presents the pseudocode for the FDIA attack. In the algorithm, the attacker performs a reconnaissance phase. The first step is to explore information related to power systems. The second step highlights how the attacker executes FDIA on power systems.
**Algorithm 1:** Attack algorithmSTEP 1: Collection of informationFrom_Node←NodeononeendofatransmissionlinesTo_Node←Nodeontheotherendofatransmissionlines$P_{d}^{i}\leftarrow Power\;demand\;at\;node\;i$$\Delta T_{L_{x}}\leftarrow load\;demand\;to\;falsify$$\Omega_{net}\leftarrow Set\;of\;all\;nodes$$T_{Lx}\leftarrow Target$STEP 2: Attack execution$P_{d}^{From\_Node_{T_{Lx}}}=P_{d}^{From\_Node_{T_{LX}}}-\Delta T_{Lx}$$P_{d}^{To\_Node_{T_{Lx}}}=P_{d}^{To\_Node_{T_{Lx}}}+\Delta T_{Lx}$$\sum_{i=1}^{\Omega_{net}}P_{d\;(\mathrm{Before}\;\mathrm{attack})}^{i}=\sum_{i=1}^{\Omega_{net}}P_{d\;(\mathrm{After}\;\mathrm{attack})}^{i}$

### 3.3. FDIA Targets in the Power Systems

In power systems, the potential and most critical FDIA targets are load demand (power demand), power supply (generation), network state, and electricity pricing [56]. Load demand is one of the most critical attack targets as it can easily increase both the cost of power generation and customer cost of power consumption. Moreover, FDIA on energy demand can render the power system unstable. This can happen when the falsified energy demand is far more than the actual energy required by the customers, or when the falsified energy demand is much less than the actual energy required by the customers. The amount of actual power generation is provided by the power generation nodes within the power systems network. FDIA on power generation can lead to starvation of the load demand nodes or an increase in the cost of power generation. Power systems network parameters, such as the transmission line capacity and the network topology, are also potential targets for false data injection attacks. An attacker can isolate a node within the power system network. This can mislead the power distribution process of the power system. In addition, electricity pricing is another potential target for FDIA. False electricity pricing would lead to the loss of revenue to the power generation companies.

### 3.4. Minimum Effort FDIA

FDIA can be executed with minimum effort from the attackers’ perspective. The effort is in terms of the number of buses to compromise and the amount of false information to be injected into the attack targets. Therefore, the objective of the attacker is to compromise the minimum number of buses and inject a minimum amount of false information. To attack the system with the minimum effort from the attacker’s perspective, the transmission line with the smallest difference between power limitation and actual power flow is selected as the target line. The nodes that connect this particular line are selected as nodes to compromise [1]. Algorithm 2 presents the pseudocode for the FDIA with minimum effort from the attackers’ perspective.
**Algorithm 2:** Minimum effort attack algorithm
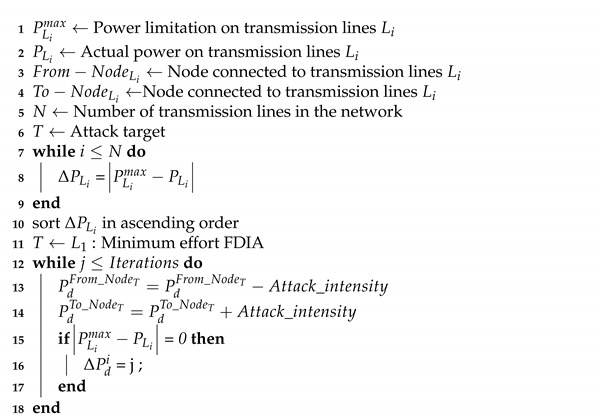


To evaluate the performance of the FDIA with a minimum effort from the attackers’ perspective, we conducted a series of experiments on the standard IEEE 6, 9, 30, and 118 bus systems. We found that there is no significant increase in the power generation cost when different bus systems are attacked with minimum effort from the attackers’ perspective. As shown in Table 3, the attack deviates the minimum cost of power generation by up to 0.41%, 0.008%, 0.0004% for the IEEE 6-bus, 9-bus, and 118-bus, respectively. Hence, the attack with minimum effort does not have a significant impact on the cost of power generation. Therefore, we need to formulate an attack strategy that will provide a significant impact on the bus systems.

## 4. Rule-Based Detection System for FDIA

This section presents the detection and prevention system for the FDIA in power systems. We extract one primary detection feature and four secondary detection features. Both primary and secondary detection features help in detecting and preventing FDIA in power systems. First, the detection and prevention mechanism checks the primary detection feature. If the primary detection feature is true, the system proceed to check the secondary detection features.

The primary detection feature is a surge in generation cost that is greater than a threshold value denoted by $C_{threshold}$. The secondary detection features are checked as follows. For the first secondary detection feature, the detection and prevention system checks if there is a change in load demand and the changes is an increase and decrease in load demand in the power systems. This is performed by comparing the present load demand value and the previous load demand value of every node in the power systems. The first secondary detection feature is donated by +/−.

For the second secondary detection feature, the detection and prevention mechanism checks if the increase in load demands happened on one node and the decrease in load demands happened on another node in the power systems network. This detection feature is donated by T-N which represent the target nodes and it is the second secondary detection feature that helps in determining if the system is under FDIA attacks or not. For the third detection feature, the detection and prevention mechanism checks if the load demand increases and decreases by the same amount on different nodes within the power systems. This secondary detection feature is donated by $\pm P_{d}$. It is the third detection feature that helps us determine if the power system is under FDIA attacks or not. Finally, for the fourth secondary detection feature, the detection and prevention mechanism checks if the change in load demand, that is increase in load demand on one node and decrease in load demand on another node happened at the same time. This detection feature is donated by δt. If all the secondary detection feature are satisified, the detection and prevention system triggers an attack alarm and reverts the system configuration to the previous normal system configuration. This prevents the impact of the FDIA on the cost of power generation and the impact on the physical components of the power systems.

To evaluate the performance of the detection and prevention system, we used the accuracy metrics TP rate, TN rate, FP rate, and FN rate. A threshold parameter $\tau_{pd}$ is used to control the accuracy metrics of the detection systems. Throughout our simulation studies, we found that the value of FP is zero for all the standard bus systems under investigation, this means that our detection mechanism never indicates the presence of an attack when there is no actual attack happening in the system. Therefore, the only important metrics that shows the performance of our detection systems are the TP, TN, and FN. To determine the sensitivity of the detection system, we find the threshold value that minimizes the number of FP. The threshold parameter is donated by $\tau_{pd}$, and it is determined by running multiple simulations on the IEEE bus systems. To prevent FDIA, after the system detected there is an attack, the system reverts the power system state and configuration to the previously known normal operation state of the system. This helps in combating the increase in generation costs incurred by the FDIA.

## 5. Results and Discussions

In this section, we first present the simulation environment and the simulation setup. Next, we present our simulation results of FDIA with minimum and unbounded effort from the attackers’ perspective. Next, we present the results of our proposed APT-based FDIA that intends to maximize the power generation cost and the detection mechanism that intends to mitigate such attacks in power systems. Last we compare our results with the previously proposed FDIA from the literature.

### 5.1. Simulation Setup

The impact of FDIA on the power systems is studied using simulation. Specifically, the impact on the cost of power generation and the impact on the operation of the generators in power systems is investigated. In addition, the attacks are simulated on the IEEE bus systems using MATPOWER MATLAB package [15]. MATPOWER is an open-source and free tool for electric power systems simulations. All simulations are carried out on a 2.3 GHz Intel Core i5 CPU and 8 GB RAM laptop.

Table 4 present the simulation parameters that include the maximum temperature and cooling rate for the simulated annealing, the population size for the genetic algorithm and particle swarm optimization algorithms, tabu list does not have a specific value because the is dynamic, the initial cost of all the standard IEEE bus systems under investigation. We chose a very high temperature and a small cooling rate in order to observe the behavior of simulated annealing when solving the optimization problem. Furthermore, we chose a population size of 50, which is enough to observe the behavior of genetic and particle warm optimization algorithms. The initial cost of 3046.03 dollars per hour, 5216.03 dollars per hour, 951.62 dollars per hour, 125,947.9 dollars per hour are the normal cost of power generation when the system is under normal operation without FDIA attacks.

$C_{threshold}$ represent the cost increase threshold which is a dynamic value. The $C_{threshold}$ unit is in dollar per hour and it the value the system can afford to lose. This cost increase threshold lead to the relationship between the generation cost of the power system and the accuracy metrics of the detection and prevention system. When the attacker falsify the load demand and the increase in generation cost in less than the $C_{threshold}$ the detection system will not be able to detect this attack because is incur negligible effect on the system. This leads to the occurrence of the FN. That is is an actual attack and the detection system detects no attack in the systems due to the negligible impact of the attack.

In the coming sections, we present the simulation results of our proposed APT-based FDIA strategy using the IEEE 6-Bus, 9-bus, IEEE 30-Bus, and IEEE 118-Bus. The results are summarized in Table 5. We compare our attack model with the attack in Reference [1] as shown in Table 6. Each simulation is repeated 10 times to get better confidence in determining the maximum increase in generation cost.

### 5.2. APT-Based FDIA on IEEE 6 Bus Systems

To evaluate the proposed FDIA, the IEEE 6-bus system is used as a test case to investigate the impact of FDIA against OPF in power systems. The system is composed of six buses with three generators at bus numbers 1, 2, and 3. Moreover, eleven transmission lines connect all buses to form a power system network. Figure 5a presents the results of attacks on 6-bus systems. From the figures, the attacker is able to achieve an up to 15.6% increase in the cost of power generation when the standard IEEE 6-bus system is attacked. Moreover, to analyze the effect of the FDIA on generators, we recorded the generator out variation of all generators in the system both before and after the attack execution. Figure 5b presents the generator output variation. As shown in the figure, the output of the generator at bus number 1 increases from 50 MW to 120.28 MW due to the FDIA attack. Therefore, the attack cause a significant impact on the generator at bus number 1. Moreover, as shown in Figure 5a, Figure 6a, Figure 7a, and Figure 8a, we observed a fluctuation when solving the optimization problem using simulated annealing. This is because the simulated annealing algorithm is a single solution-based metaheuristics, and it combines both the notion of exploring and exploiting. Exploring is the idea of visiting more search space with the hope that the algorithm will get a better solution. However, the notion of exploiting is that the algorithm is always trying to find a global optimum as quickly as possible, this can lead it to stuck at a local optimum. That is, during the hill-climbing, the algorithm always wanted to exploit.

### 5.3. APT-Based FDIA on IEEE 9-Bus System

The IEEE 9-bus system consists of nine buses with three generators at bus number 1, bus number 2, and bus number 3. Moreover, nine transmission lines connect all buses in the system to form a power system network. Figure 6a presents the results of attacks on 9-bus systems. From the figures, the attacker can achieve an up to 45.1% increase in the cost of power generation when the standard IEEE 9-bus system is attacked. Moreover, as shown in Figure 6b, the attack cause a significant impact on the generator at bus number 2 for the standard 9-bus systems. The output of the generator increases from 134.38 MW to 269.30 MW due to the FDIA attack. This the major factor that leads to an increase in power generation cost.

### 5.4. APT-Based FDIA on IEEE 30-Bus System

The 30-bus system consists of thirty buses. Moreover, these buses are connected with 41 transmission lines to form the power system network. The system also consists of six generators connected to bus numbers 1, 2, 5, 8, 11, and 13.

Figure 7a presents the results of attacks on the 30-bus systems. From the figures, the attacker can achieve an average of up to 57.7% increase in the cost of power generation when the standard IEEE 30-bus systems is attacked. As shown in Figure 7b, the attacks cause a significant impact on the generator at bus number 1. The output of the generator increases by about 70 MW due to the FDIA attack. Consequently, it leads to the increase in the fuel cost.

### 5.5. APT-Based FDIA on IEEE 118-Bus System

The 118-bus system consists of 118 buses with 54 generator buses. The buses are connected by 186 transmission lines to form the power system network. Figure 8a presents the results of attacks on the 118-bus systems. From the figures, the attacker can achieve an average of up to 71.7% increase in the cost of power generation when the standard IEEE 118-bus system is attacked. As shown in Figure 8b, the attack cause a significant impact on the generator at bus number 69. The output of the generator increases from 500.43 MW to 796.06 MW due to the FDIA attack.

### 5.6. FDIA Detection Results

In this section, we present the simulation results of our proposed detection and prevention mechanism for the IEEE 6-Bus, 9-bus, IEEE 30-Bus, and IEEE 118-Bus. The detection and prevention mechanism is simulated using MATPOWER MATLAB package [15]. All simulations are carried out on a 2.3 GHz Intel Core i5 CPU and 8 GB RAM laptop. Figure 9a, Figure 10a, Figure 11a, and Figure 12a present the generation cost both before and after deploying our detection and prevention system. The figures show that our detection and prevention system is capable of detecting and blocking the FDIA and take the systems back to the normal operation state. Accuracy metrics are used to evaluate the performance of the developed rule-based FDIA detection and prevention system.

Figure 9b, Figure 10b, Figure 11b, and Figure 12b present the percentage occurrence of the accuracy metrics at a specific load demand threshold value. The accuracy metrics are the True Positive (TP) rate, True Negative (TN) rate, False Positive (FP) rate, and False Negative (PN) rate. We observed that the rate of the accuracy metric True Positive (TP) is greater than 90% for the all the IEEE standard 6, 9, 30, and 118 bus systems. This shows the good performance of our detection and prevention system. It also shows that the system almost always accurately detect the presence of an attack when there is an actual attack on the power systems. We conducted several simulations to evaluate the sensitivity of the load demand threshold $\tau -P_{d}$. In addition, we observed that the rate of the FP is always zero (0) on all the bus systems under investigation. This also shows the good performance of our detection and prevention system. This is because the accuracy metric FP indicates when there is no attack and a detection system detects an attack in the system. Therefore, when the value of FP is zero (0), this means that the detection and prevention never indicate the presence of an attack when there is no actual attack on the system. In addition, the reason for the FP becoming zero (0) in our system is because there is no presence of noise during our simulation.

As shown in Figure 13a,b and Figure 14a,b, we found that the False Negatives (FN) rates become significantly low as the value of load demand threshold $\tau_{pd}$ increases. Specifically, the FN rates approaches 0 when $\tau_{pd}>130$ MW, $\tau_{pd}>300$ MW, $\tau_{pd}>150$ MW, and $\tau_{pd}>1200$ MW for the standard IEEE 6-bus, 9-bus, 30-bus, and 118-bus systems, respectively.

Figure 15a,b present the comparison of the percentage of occurrence of the accuracy metrics in our proposed detection and prevention system and the detection mechanism in Reference [1] from the literature. From the figure the false positive rates of the detection mechanism proposed in in in Reference [1] becomes significantly low as the value of load active power threshold approaches 40 MW on both IEEE 30, and 118 bus systems. This means that their system performs better when the threshold value is greater than 40 MW. On the other hand, our proposed detection and prevention system performs better when the threshold values approaches 150 MW and 1200 MW on the standard IEEE 30 and 118 bus systems, respectively.

## 6. Conclusions and Future Work

OPF is one of the most important modules in power systems that ensure the optimal operation of power systems. In this paper, we analyzed the impact of FDIA against OPF in power systems. Then, we, namely, simulated annealing, genetic algorithm, and particle swarm optimization. OPF is one of the most important modules in power systems that ensure the optimal operation of power systems. In addition, we introduced a novel APT-based attack strategy with the objective to maximize the operation cost of power generation. We modeled the attack strategy as an optimization problem. We used metaheuristic algorithms, namely: genetic algorithm (GA), simulated annealing (SA) algorithm, tabu search (TS), and particle swarm optimization (PSO) to solve the proposed optimization problem and obtain the attack plan. We conducted several simulation scenarios and observed that our proposed attack strategy can increase the power generation cost by up to 15.6%, 45.1%, 60.12%, and 74.02% on the 6-bus, 9-bus, 30-bus, and 118-bus systems, respectively. In addition, we observed a fluctuation when solving the optimization problem using simulated annealing. This is expected, because the simulated annealing algorithm is a single solution-based metaheuristics, and it combines both the notion of exploring and exploiting. That is, during the hill-climbing, the algorithm always wanted to exploit. In addition, the attacks cause significant impact on the generators performance on the bus number 1, 2, and 69 for the 6-bus, 9-bus, 30-bus, and 118-bus systems, respectively. We developed a rule-based FDIA detection and prevention system with primary and secondary detection features. The system is able to detect and Prevent FDIA in power systems. Moreover, accuracy metrics are used to evaluate the performance of the developed rule-based FDIA detection and prevention system. We found that the False Negatives (FN) rates become significantly low as the value of load demand threshold $\tau_{pd}$ increases. Specifically, the FN rates approaches 0 when $\tau_{pd}>130$ MW, $\tau_{pd}>300$ MW, $\tau_{pd}>150$ MW, and $\tau_{pd}>1200$ MW for the standard IEEE 6-bus, 9-bus, 30-bus, and 118-bus systems, respectively. In the future, we intend to use other metaheuristics algorithms to check the possibility of improving our results. Finally, the main limitation of this work is the use of only DC optimal power flow without considering the AC optimal power flow. This is because the DC optimal power flow is the most extreme approximation of the power flow. Another reason for choosing DC over AC is due to problems related to the AC optimal power flow. Although AC optimal power flow solutions should be more accurate than DC optimal power flow, its convergence problem is quite severe. The main challenge faced in this study is the challenge related to the optimal power flow learning curve. That is, the steep learning curve of the OPF before the final solutions are obtained. Therefore, early solutions of the OPF must be carefully examined.

## Figures and Tables

**Figure 1 sensors-21-02478-f001:**
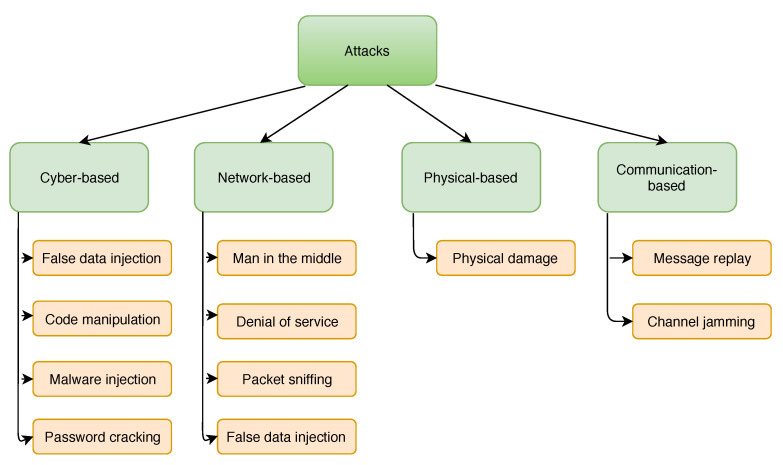
Classifications of attacks in energy industry [13].

**Figure 2 sensors-21-02478-f002:**
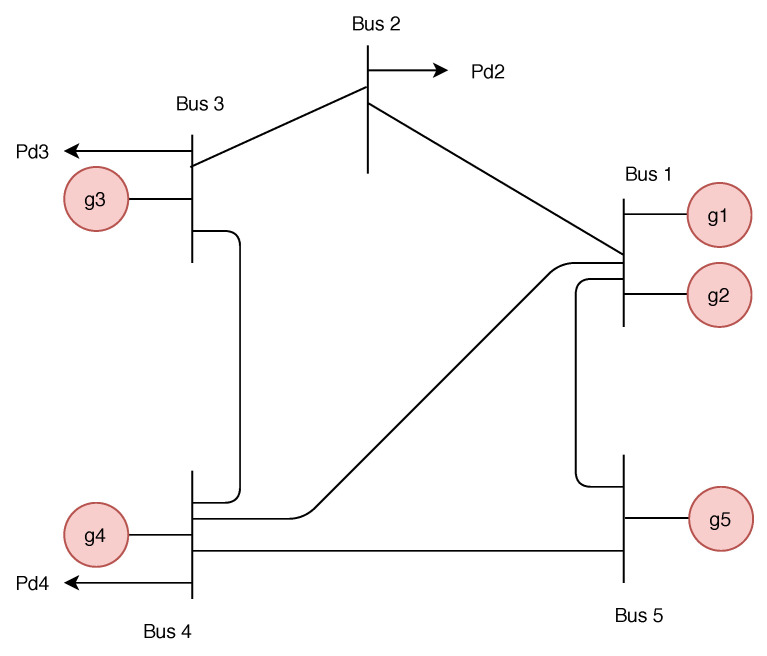
Schematic diagram of a 5-bus system.

**Figure 3 sensors-21-02478-f003:**
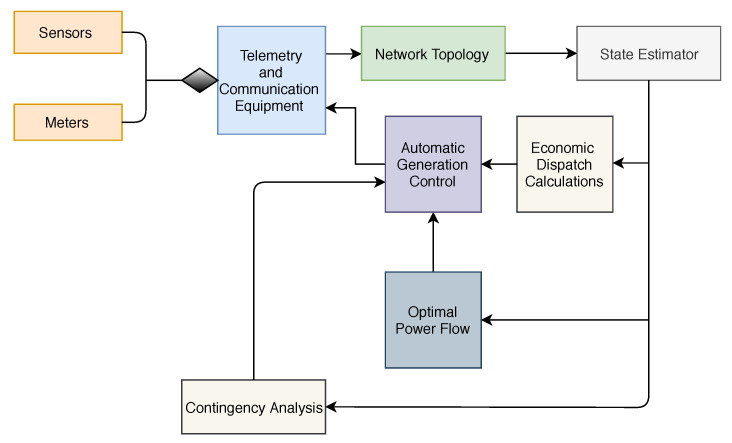
System architecture of the energy management system [1].

**Figure 4 sensors-21-02478-f004:**
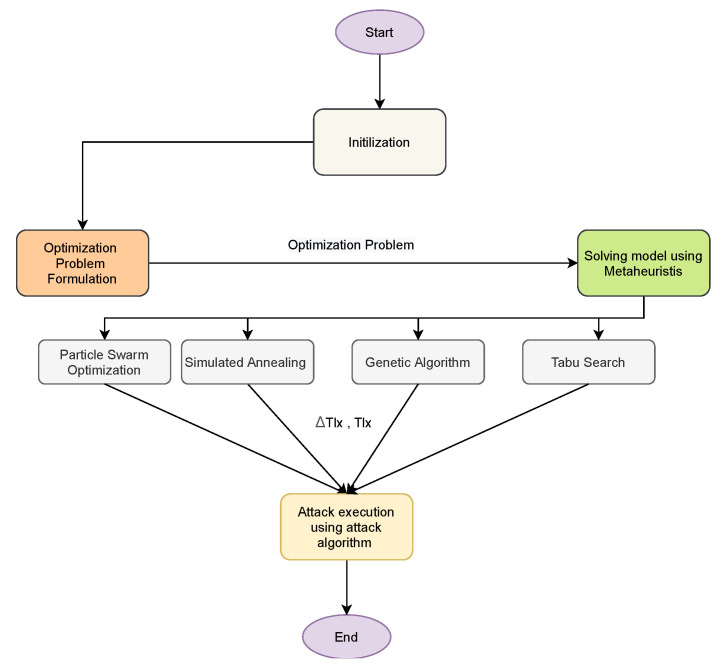
Flowchart of the Attack plan.

**Figure 5 sensors-21-02478-f005:**
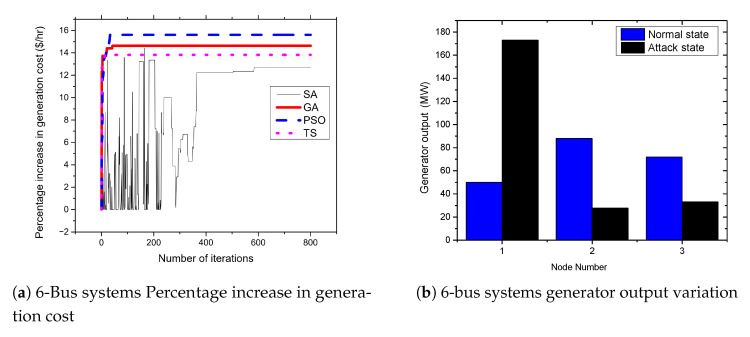
6-bus systems increase in generation cost and generator output variation.

**Figure 6 sensors-21-02478-f006:**
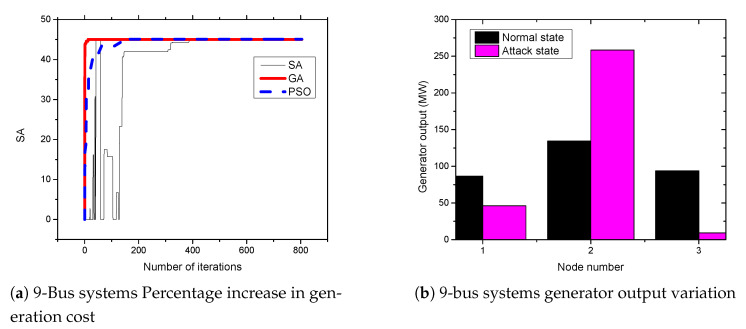
9-bus systems increase in generation cost and generator output variation.

**Figure 7 sensors-21-02478-f007:**
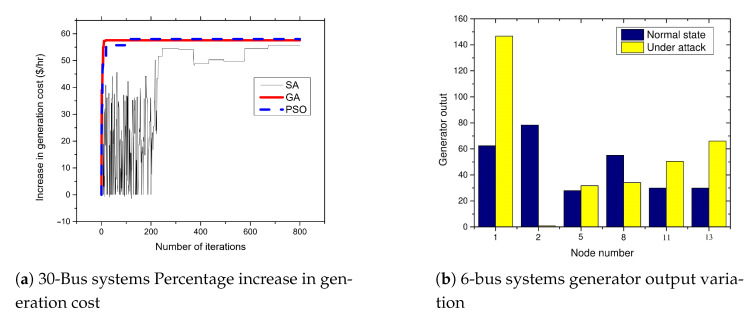
30-bus systems increase in generation cost and generator output variation.

**Figure 8 sensors-21-02478-f008:**
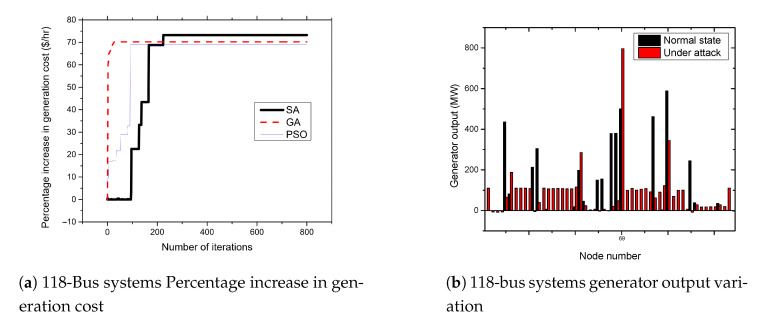
118-bus systems increase in generation cost and generator output variation.

**Figure 9 sensors-21-02478-f009:**
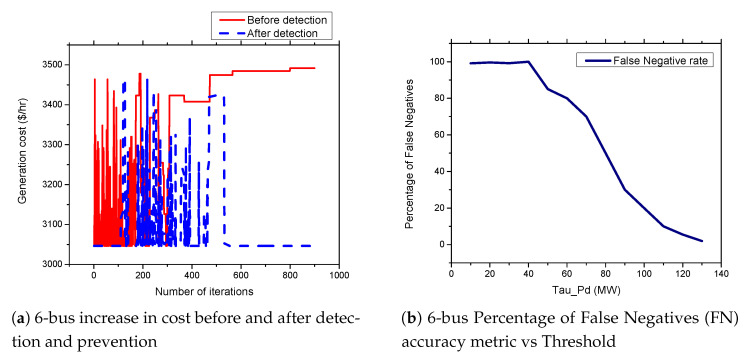
6-bus systems increase in cost before and after detection and prevention and percentage of False Negatives (FN) accuracy metric vs. Threshold.

**Figure 10 sensors-21-02478-f010:**
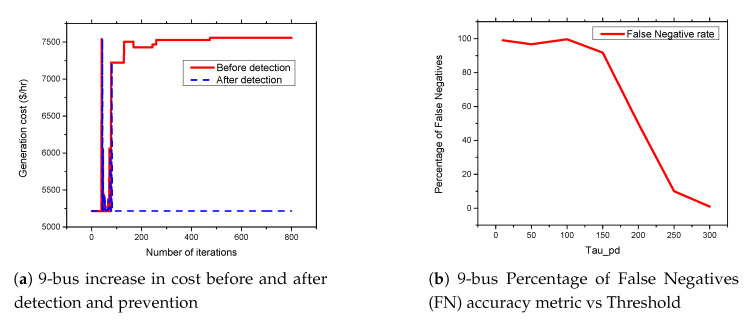
9-bus systems increase in cost before and after detection and prevention and percentage of False Negatives (FN) accuracy metric vs. Threshold.

**Figure 11 sensors-21-02478-f011:**
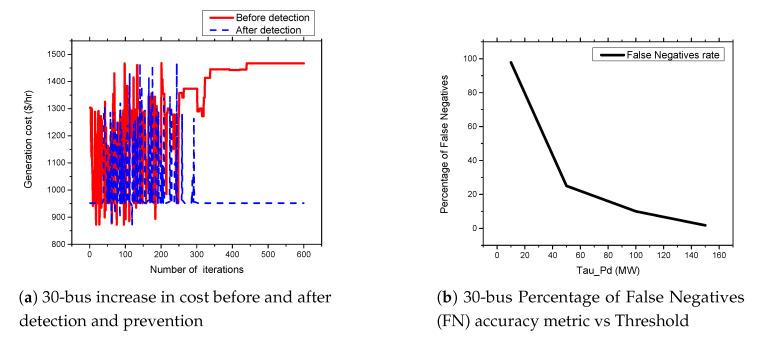
30-bus systems increase in cost before and after detection and prevention and percentage of False Negatives (FN) accuracy metric vs. Threshold.

**Figure 12 sensors-21-02478-f012:**
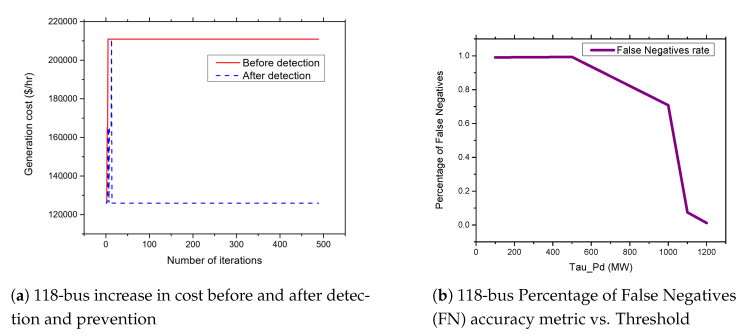
118-bus systems increase in cost before and after detection and prevention and percentage of False Negatives (FN) accuracy metric vs. Threshold.

**Figure 13 sensors-21-02478-f013:**
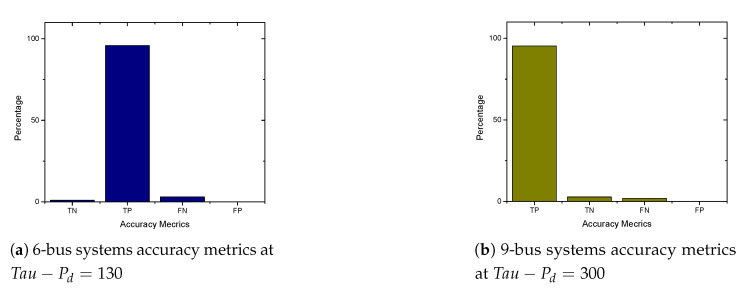
6-bus and 9-bus systems accuracy metrics.

**Figure 14 sensors-21-02478-f014:**
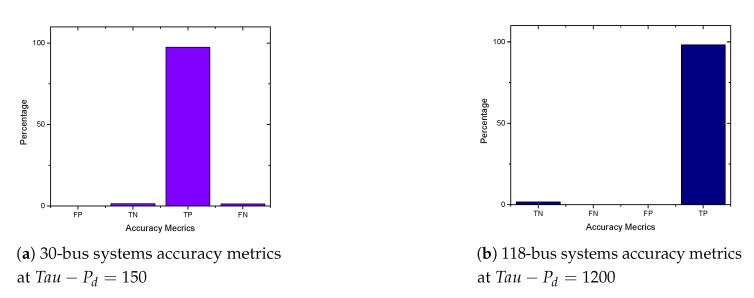
30-bus and 118 bus systems accuracy metrics.

**Figure 15 sensors-21-02478-f015:**
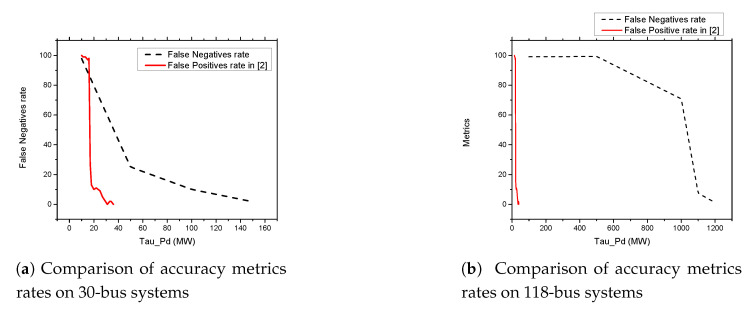
Comparison of accuracy metrics rates on 30 and 118-bus systems.

**Table 1 sensors-21-02478-t001:** Examples of cyber attacks on power systems [13].

Attacks	Year	Region	Consequences
False Data Injection Attacks	2015	Kyiv, Ukraine	Several hours of power outage (blackout). Affecting about 225,000 customers.
	2008	Turkey	Explosion of oil pipeline in which 30,000 barrels of oil is spilled in water.
	2007	Idoha National Lab, USA	Generator exploded.
Code Manipulation	1999	Bellingham, USA	Three people were killed by a huge fireball and many others were injured.
	1982	Russia	Explosion of 3 kilotons of Trinitrotoluene (TNT)
Malware Injection	2012	Saudi Arabia, and Qatar	Energy distribution, and energy generation are affected.
	2003	Ohio, USA	System shutdown for 5 h.

**Table 2 sensors-21-02478-t002:** Modeling parameters.

Notation	Notation Definition
*R*	Set of all From nodes
*L*	Set of all To nodes
$\Omega_{net}$	Set of all nodes (buses) in the network
$\Omega_{d}$	Set of all nodes (buses) in which a demand/load is connected
$\Omega_{g}$	Set of all nodes (buses) in which a generator is connected
$\Omega_{L_{i}}$	Set of all lines connected to the *i*th node (bus)
$P_{g_{i}}$	Generator output of *i*th generator
$P_{g}^{min}$	Lower bound of the generator output
$P_{g}^{max}$	upper bound of the generator output
$P_{d_{i}}$	Load/demand at node (bus) *i*
$\delta_{i}$	Voltage phase angle of node (bus) *i*
$P_{ij}$	Power flow between bus *i* and *j*
$P_{ij}^{max}$	Upper bound of the Power flow between node (bus) *i* and *j*
$B_{ij}$	Line admittance between node (bus) *i* and *j*
$C_{i}\left(P_{g_{i}}\right)$	Cost function of the generator at *i*th node (bus)
$T_{Lx}$	Target transmission lines
$\Delta T_{Lx}$	False load demand value
α	Control parameter

**Table 3 sensors-21-02478-t003:** Generation cost increase under minimum effort False Data Injection Attacks (FDIA) ($/hr).

Bus-Systems	Normal Systems State	Minimum Effort Attack
6-bus systems	3046.413	3059.031
		(0.41%)
9-bus systems	5216.026	5216.477
		(0.008%)
30-bus systems	951.62	951.62
		(0%)
118-bus systems	125,947.88	125,948.37
		(0.0004%)

**Table 4 sensors-21-02478-t004:** Simulation parameters.

Notation	Value	Notation Definition
$T_{max}$	1000	Maximum temperature
$T_{min}$	0	Minimum temperature
C	0.99	Cooling rate
P	50	Population size
$\tau -P_{d}$	N/A	Dynamic false load demand
$T_{list}$	N/A	Dynamic tabu list
$C_{init}$	3046.03	6-bus systems initial cost
$C_{init}$	5216.03	9-bus systems initial cost
$C_{init}$	951.62	30-bus systems initial cost
$C_{init}$	125,947.9	118-bus systems initial cost
$C_{threshold}$	N/A	Dynamic cost increase threshold

**Table 5 sensors-21-02478-t005:** Minimum, average, and Maximum cost increase under attack on different bus systems.

Bus Systems	Metaheuristics	Min. Cost (*$*/hr)	Avg. Cost (*$*/hr)	Max. Cost ($/hr)
6-Bus System	SA	3484.9	3490.03	3491.73
	GA	3491.72	3491.73	3491.74
	PSO	**3492.21**	**3501.80**	**3521.65**
	TA	3455.64	3462.456	3467
9-Bus System	SA	7527.79	7555.77	7564.83
	GA	**7564.81**	**7564.82**	7564.83
	PSO	7478.89	7541.73	**7568.70**
	TA	7478.89	7496.945	7515
30-Bus System	SA	1481.58	1499.982	1517.57
	GA	**1494.56**	**1509.62**	1517.57
	PSO	1492.11	1505.99	**1523.75**
	TA	1469	1469	1470
118-Bus System	SA	**212,166.41**	**216,312.38**	**219,180.72**
	GA	212,108.16	213,703.66	214,387.45
	PSO	211,535.02	215,681.64	218,954.18
	TA	211,000	211,900	212,500

**Table 6 sensors-21-02478-t006:** Comparison between our work and the work in Reference [1].

Bus Systems	Normal	Min. Effort [1]	APT-Based
6-bus systems	3046.41	3256.37	3521.65
		(6.89%)	(15.6%)
9-bus systems	5216.03	6652.88	7568.70
		(27.55%)	(45.1%)
30-Bus systems	951.62	1034.49	1523.75
		(8.7%)	(60.12%)
118-bus systems	125,947.88	132,697.88	219,180.72
		(5.36%)	(74.02%)

## Data Availability

Not applicable.

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
