# Peer review of "Rule-Based Detection of False Data Injections Attacks against Optimal Power Flow in Power Systems"

_sensors, 2021, doi:10.3390/s21072478_

Round 1

Reviewer 1 Report

The authors have presented an interesting research on FDIA. A number of case studies have comprehensively justified and explained. Comments are below. 1. Could you further justify equation 1? 2. On page 14, could you confirm the pseudo code is correct? 3. Could you also compare the computation efforts for different optimization tools?

Author Response

We would like to thank the reviewer for their valuable comments to improve the manuscript. We have addressed all comments. The details of our response are below. The paper has been thoroughly revised. All the comments of the reviewers have been carefully considered and addressed.

Comments of Reviewer # 1

 Response

1.

Could you further justify equation 1?

Thank you for your comment. Equation 1 presents the model of the operational cost of the generators in terms of fuel consumption. Based on the literature, the cost of a particular generator  is modeled as a quadratic cost function provided in Equation 1 where , , and  are the cost coefficients related to the fuel cost. Moreover, the cost coefficients are determined by the physical properties of the generators.

In this paper, the main objective of our attack is to maximize the cost of power generation in the power system. Therefore, we have used the fuel cost of the generation as the quadratic objective function of the optimal power flow problem formulation.

The description of Equation 1 has been also incorporated in Secution 2.5.

2.

On page 14, could you confirm the pseudo code is correct?

Thank you for your comment. The pseudo-code on page 14 was scrambled due to conflict in the latex packages. The pseudo-code on page 14 has been corrected.

3.

Could you also compare the computation

efforts for different optimization tools?

Thank you for your comment. In Figures 5a, 6a, 7a, and 8a, we have presented the algorithms convergence in terms of the number of iterations of each of the optimizations tools we have used in this paper.  

As presented in the figures, all the metaheuristics used to solve the formulated optimization problem have relatively different convergence rates.

Reviewer 2 Report

The subject of this study is of interest both in power systems and cybersecurity. The authors are invited to address the following comments to meet the standards of publication:

  1. The abstract should be concise. Please shorten the abstract to include more informative descriptions.
  2. Please provide a table for acronyms to increase the readability of the manuscript. 
  3. In the current literature review, it is difficult to point out the main contributions of the study.
  4. Please avoid lump references, e.g. line 33. All the references should be explained individually. 
  5. Line 430: what are Equations 3.2 and 3.5? The same problem for other equation captions in the body text.
  6. The objective of optimal power flow can be defined as power generation cost, power loss in transmission lines, etc. In Eq.(1), the power generation cost is addressed. what about the cost of power loss? is it possible to study the impact of the suggested approach on power loss cost?
  7. One of the main aims of Sensors Journal is to introduce the industrial applications of the proposed approach. Could you please explain the application of your approach in real power systems?
  8. In addition to the applications of your approach, the authors should mention the main challenges and limitations of the current study. I saw barely any suggestions in the Conclusion Section. Please provide some clear explanations.

Author Response

We would like to thank the reviewer for their valuable comments to improve the manuscript. We have addressed all comments. The details of our response are below. The paper has been thoroughly revised. All the comments of the reviewers have been carefully considered and addressed

Comments of Reviewer # 2

 Response

1.

The abstract should be concise. Please shorten the abstract to include more informative

descriptions.

Thank you for your comment. The abstract has been carefully reviewed and updated accordingly.

2.

Please provide a table for acronyms to increase the readability of the manuscript.

Thank you for your comment. Table 9 has been added, which contains acronyms and their definitions for easier readability of the manuscript.

3.

In the current literature review, it is difficult to point out the main contributions of the study.

Thank you for your comment. The main contribution of this paper is as the following. First, we provide a detailed analysis of the impact of False Data Injection Attack (FDIA) on the power systems. Second, we have proposed an Advanced Persistence Threat (APT) based FDIA that intends to maximize the negative impact on the cost of power generation and the physical component of the power systems. Third, we formulate the attack model as an optimization problem to maximize the cost of power generation. The main difference between our proposed APT-based FDIA and the proposed FDIA from the literature is we have used metaheuristics techniques, namely, Genetic Algorithm (GA), 48 Simulated Annealing (SA), and Particle Swarm Optimization (PSO) to conduct the attack. Then, we have also proposed a rule-based detection mechanism to mitigate FDIA in power systems. Finally, we evaluate the attacks and the detection mechanisms by running simulations on the standard IEEE bus systems using MATPOWER in 50 MATLAB.

4.

Please avoid lump references, e.g. line 33. All the references should be explained individually.

Thank you for your comment. The lump references in Section 1 have been carefully reviewed and updated. Lump references have also been avoided throughout the paper.

5.

Line 430: what are Equations 3.2 and 3.5? The same problem for other equation captions in the

body text.

Thank you for your comment. We have carefully reviewed the captions of the equations and updated them accordingly. Equations 3.2 to 3.5 are now Equations 2,3,4, and 5.

6.

The objective of optimal power flow can be defined as power generation cost, power loss in transmission lines, etc. In Eq.(1), the power generation cost is addressed. what about the cost of power loss? is it possible to study the impact of the suggested approach on power loss cost?

Thank you for your comment. In this paper, we have assumed that the losses in the transmission lines are all zero. Therefore, the sum of the active power output of all generators is equals to the sum of the load active power of all nodes in the power systems network.

7

One of the main aims of Sensors Journal is to introduce the industrial applications of the

 proposed approach. Could you please explain the application of your approach in real power systems?

The proposed attacks and defensive strategies can be implemented in real power systems. One hand, adversaries can physically compromise measurement equipment such as electric meters and sensors. On the other hand, attackers can also intercept data packets in the power system network while they are transferred to the control center. Moreover, attackers can modify the database of the control center through unauthorized access. In this paper, we focus on the unauthorized falsification of data will mislead the operation of the power systems which leads to an increase in the generation cost.

The proposed defensive mechanism can be implemented in real power systems control. This will enhance decision-making and help in detecting and preventing false data injection attacks against real power systems. In addition, it is important to note that the standard IEEE  bus systems are a simple approximation of the real power systems.  Hence, we have used them to evaluate the performance of the attacks and defensive mechanism.

8

In addition to the applications of your approach, the authors should mention the main challenges and limitations of the current study. I saw barely any suggestions in the Conclusion Section. Please provide some clear explanations.

The main limitation of this research is the use of only DC optimal power flow without considering the AC optimal power flow. This is because the DC optimal power flow is the most extreme approximation of the power flow. Another reason for choosing DC over AC is due to problems related to the AC optimal power flow. Although AC optimal power flow solutions should be more accurate than DC optimal power flow, its convergence problem is quite severe. The main challenge faced in this study is the challenge related to the optimal power flow learning curve. That is, the steep learning curve of the OPF before the final solutions are obtained. Therefore, early solutions of the OPF must be carefully examined. 

Reviewer 3 Report

The paper is an interesting contribution to false data injections attacks aginst optimal power flow in power systems. The paper shows several weaknesses and strenghs. The general characteristic is the lack of information about the details of proposed algorithms.

Weaknesses:

  • Figure 4, page 11 needs more details, tools, data flow, algorithms, in the present form does not make any contribution.
  • In the section 3.1.1 pag. 10-12, several computational intelligence are described. However, the description is incompleted, for example, how de information is converted to genes? what is the fitness function of each algorithm? how do you calculate the cost of new possible solution? In PSO what is the equivalent of velocity or position of particle? In the simulated annealing what is the parameter which represents the temperature?
  • The equation reference has some mistakes, for example, page 13, line 428 references equations 3.1-3.5.
  • Algorithm 1, page 14, should be corrected.
  • The section 4, I think which is the focus of the paper, but is incompleted, too. How many rules are implemented? what is the architecture or tehcnology-based?
  • Conclusion and future work. If I understand well, the authors used computational intelligence to simulate the attackers and a rule-based system to simulate the defender. I need to know if the defender has any dynamic characteristic (rules used to be static), which allow to adapt to the attackers. So, I do not know the naumber of rules or the module architecture and tchnology-based, I think it is important to discuss about that.

Strengths:

  • Interesting topic.
  • Good state of art.

Author Response

We would like to thank the reviewers for their valuable comments to improve the manuscript. We have addressed all comments. The details of our response are below. The paper has been thoroughly revised. All the comments of the reviewers have been carefully considered and addressed.

Comments of Reviewer # 3

Authors Response

1.

Figure 4, page 11 needs more details, tools, data flow, algorithms, in the present form does not make any contribution.

Figure 4 is carefully reviewed and updated. Moreover, the figure highlights a simple flowchart of the attack plan.First, we start with the FDIA Modeling, the output of the FDIA model is the formulated optimization problem. Next, the formulated optimization problem is solved using metaheuristics. The output of the solution is the attack targets and the amount of load demand that is feasible to falsify. This provides the attack plan for the attacker to be able to execute FDIA in power systems to maximize the impact of the attack on the power systems. The adversary performs a reconnaissance. The first step is to explore information related to power systems. In the second step, the adversary falsifies the load demand of the target nodes to executes FDIA.

2.

In the section 3.1.1 page. 10-12, several computational intelligence are described. However, the description is incompleted, for example, how de information is converted to genes? what is the fitness function of each algorithm? how do you calculate the cost of new possible solution? In PSO what is the equivalent of velocity or position of particle? In the simulated annealing what is the parameter which represents the temperature?

Thank you for your comment. Section 3.1.1 has been carefully reviewed and updated accordingly.

The cost of new possible solutions is the value of the objective function of the optimal power flow in dollars per hour. This is calculated with the help of MATPOWER simulator which is an open-source and free tool for electric power systems simulations. Table 6 presents the simulation parameters. Moreover, the objective function of the optimal power flow is the fitness function of each of the metaheuristics used in this study. In simulated annealing, we have used a maximum temperature of 1000 and a minimum temperature of 0 with a cooling rate so that we have a sufficient number of iterations for the algorithm to converge and provide a better solution. In genetic algorithm, the chromosomes are formed as a combination of the value of the target and the value of the load demand to falsify. That is, the set of all possible attack targets and the possible load demand values to falsify. A random uniform distribution function is used to generate the initial possible values of load demand to falsify and the possible attack targets. In particle swarm optimization, each particle represents a candidate solution of a target transmission line to attack and the value of load demand to falsify. This candidate solution changes at random and moves toward a value that provides the optimal solution. The equivalence of the positions of the particles is the objective function solutions of every particle. The equivalence of the velocity is the rate of change of the feasible load demand values to falsify to archive an optimal solution.

3.

The equation reference has some mistakes, for example, page 13, line 428 references equations

3.1-3.5.

Thank you for your comment. We have carefully reviewed the captions of the equations and updated them accordingly.

4.

Algorithm 1, page 14, should be corrected.

Thank you for your comment. The algorithm has been corrected.

5.

The section 4, I think which is the focus of the paper, but is incompleted, too. How many rules are implemented? what is the architecture or tehcnology-based?

The main focus of the paper can be broadly viewed as two parts. The first part is the study of the FDIA attacks strategy and the second part to the study of the detection mechanism. The rules implemented in the rule-based detection mechanism are presented as the primary and secondary detection futures in section 4.  The futures include the surge in generation cost, change in the load demand, at what nodes are affected by the changes.

6.

Conclusion and future work. If I understand well, the authors used computational intelligence to simulate the attackers and a rule-based system to simulate the defender. I need to know if the defender has any dynamic characteristic (rules used to be static), which allow to adapt to the attackers. So, I do not know the naumber of rules or th module architecture and tchnology-based, I think it is important to discuss about that.

Thank you for your comment, the characteristics of the rule-based defending mechanism have been discussed in section 4. Moreover, the rules are the primary and the secondary detection features a discussed in section 4.

Round 2

Reviewer 2 Report

The authors have addressed my comments accordingly. 

Reviewer 3 Report

The authors satisfied all recommendations.